# Multi-Horizon Air Pollution Forecasting with Deep Neural Networks

**DOI:** 10.3390/s21041235

**Published:** 2021-02-10

**Authors:** Mirche Arsov, Eftim Zdravevski, Petre Lameski, Roberto Corizzo, Nikola Koteli, Sasho Gramatikov, Kosta Mitreski, Vladimir Trajkovik

**Affiliations:** 1Faculty of Computer Science and Engineering, Ss. Cyril and Methodius University, 1000 Skopje, North Macedonia; mirche.arsov@gmail.com (M.A.); petre.lameski@finki.ukim.mk (P.L.); nikola.koteli@hotmail.com (N.K.); sasho.gramatikov@finki.ukim.mk (S.G.); kosta.mitreski@finki.ukim.mk (K.M.); vladimir.trajkovik@finki.ukim.mk (V.T.); 2Department of Computer Science, American University, Washington, DC 20016, USA; rcorizzo@american.edu

**Keywords:** RNN, LSTM, convolutional networks, deep learning, air pollution

## Abstract

Air pollution is a global problem, especially in urban areas where the population density is very high due to the diverse pollutant sources such as vehicles, industrial plants, buildings, and waste. North Macedonia, as a developing country, has a serious problem with air pollution. The problem is highly present in its capital city, Skopje, where air pollution places it consistently within the top 10 cities in the world during the winter months. In this work, we propose using Recurrent Neural Network (RNN) models with long short-term memory units to predict the level of PM10 particles at 6, 12, and 24 h in the future. We employ historical air quality measurement data from sensors placed at multiple locations in Skopje and meteorological conditions such as temperature and humidity. We compare different deep learning models’ performance to an Auto-regressive Integrated Moving Average (ARIMA) model. The obtained results show that the proposed models consistently outperform the baseline model and can be successfully employed for air pollution prediction. Ultimately, we demonstrate that these models can help decision-makers and local authorities better manage the air pollution consequences by taking proactive measures.

## 1. Introduction

Air pollution is a wide spread problem [1], contributing to seven million deaths a year. About 92% of the world’s population is breathing toxic air, and it is estimated that 70% of the world’s population will live in urban centers by 2050 [2]. Per a World Health Organization report [3], air pollution is the leading cause of death for children under the age of 15 with about 600,000 deaths every year, according to a report done by the World Health Organization [3]. The financial impact of premature deaths due to air pollution is about $5 trillion in welfare losses worldwide [4]. For these reasons, efficient solutions are required to monitor and predict air pollution.

Some studies show that air pollution increases the incidence of respiratory diseases in areas with a high concentration of air pollutants such as PM2.5 and PM10 particles [5]. The authors in [6] gave a review of the challenges humanity faces with the negative impact of the particles on human health. In recent years, as the Internet of Things paradigm has become popular, sensory arrays have been placed in urban areas to collect vast amounts of data. In turn, cloud computing technologies are being used to analyze and detect emerging patterns [7] and facilitate near real-time monitoring and visualization. Support vector machine methods and artificial neural networks have been used for a variety of prediction tasks, starting from business and financial tasks [8], to a variety of ecological problems [9]. Even more beneficial are predictive air pollution systems because they can help governments employ smarter solutions and preventive measures to address air quality problems. For this reason, tracking and predicting air pollution has become a necessity in every modern urban society. The ability to predict air quality and to know when the air quality will exceed the threshold of becoming hazardous is essential for managing air pollution. It gives the authorities a tool to understand when to take preventive measures such as traffic reduction policies, the closure of public venues including schools, and recommendations to limit exposure for sensitive people [10].

The authors of [11] showed that ambient air quality data can be modeled as stochastic time series, which allows building models that can predict future values. Statistical learning models, machine learning, and deep learning models have been applied for extracting patterns directly from the input data, learning from the data distribution [12]. There are other approaches for successful forecasting over large multi-sensor data sets, using sliding window-based feature extraction and feature subset ensemble selection [13,14]. These approaches show that it is possible to use short-term predictions of dangerous methane concentrations in coal mines to take adequate measures to prevent the mines from reaching the hazardous thresholds. The same approach can be used in cities and other areas. Taking preventive measures could decrease the influence on the air quality upfront and prevent or minimize the citizens’ exposure to air with hazardous pollution [15,16].

In this article, we use air quality measurements and combine them with meteorological data to predict the air pollution in the Skopje city area for multiple time horizons of 6, 12, or 24 h. This work’s main contribution is that we combine various data sensor sources available in Skopje’s area to increase the accuracy of the predictions. Furthermore, we leverage the data from the meteorological stations and combine them with the historical data from the various air pollutants measured in geographically distant places within the city area. We evaluate different architectures based on Long Short-Term Memory (LSTM) networks and Convolutional Neural Networks (CNNs) and compare them with Auto-Regressive Integrated Moving Average (ARIMA) models.

The article is organized as follows. In Section 2, we review recent relevant works on air pollution. In Section 3, we describe the evaluated architectures and the dataset. After that, in Section 4, we show the results from the experiments, and in Section 5, we discuss and analyze the results. Finally, Section 6 concludes the paper.

## 2. Related Work

Recently, the air pollution prediction problem has been tackled with many different models considering traditional and deep learning approaches. They mainly utilize measurements of multiple pollutants, such as Particulate Matter (PM)—PM2.5 and PM10—and gaseous species (i.e., NO_2_, CO, O_3_, and SO_2_) collected from sensors at specific times and locations. The meteorological information of current and forecasted parameters like humidity, temperature, wind speed, and rainfall have also been integrated [17,18].

Predicting the concentration of PM10 for different hours by using three different step-wise Multiple Linear Regression (MLR) models was proposed in [19]. In [20], the authors forecast the concentration of different air pollutants for the current day and the subsequent four days in a highly polluted region utilizing Artificial Neural Networks (ANNs) with Real-Time Correction (RTC).

In [21], a Fuzzy Time Series Markov chain (FTSMC) model based on a grid method with an optimal number of partitions was used to predict the daily Air Pollution Index (API). A hybrid multi-resolution method that utilizes high resolution (1 h) and low resolution (1 day) data as the input and generates low-resolution PM2.5 concentrations was shown in [22].

Other popular approaches for the estimation of air pollution combine different machine learning algorithms. Some such approaches are the forecast of particulate matter concentration in atmospheric air using cross-validation evaluation by linear regression, random forests, gradient boosting, K-nearest neighbors, MLP, and CART decision trees, as proposed in [23]. In [24], the predictive value of the Weather Research and Forecasting (WRF) model was used as the input for a municipal atmospheric pollutant response model, which was based on the random forest algorithm. In [25], the authors forecast the API by adopting trigonometric regressors, Box-Cox transformation, ANNs, ARIMA, and Fuzzy Time Series (FTS).

The use of ARIMA time series models for forecasting air quality every month was presented in [26]. The authors in [27] went a step further in the use of the ARIMA models for creating a forecasting model by combining them with Empirical Mode Decomposition (EMD) and a Non-linear Auto-Regressive Neural Network (NARNN). In [28], a periodogram-based test was used to examine the periodic and seasonal components of PM10 time series, which were modeled within a trigonometric harmonic regression setup and used to forecast future values with superior accuracy compared to ARIMA.

Another class of approaches is represented by those that adopt LSTM neural networks [29]. LSTM is an artificial Recurrent Neural Network (RNN) [30] architecture used in deep learning. LSTMs are considered a typical architecture of neural networks for sequences and lists due to their chain-like nature. LSTMs have been successfully applied in forecasting tasks in a variety of domains, such as financial time series [31] and sensory data [32,33,34,35,36]. An alternative for auto-regressive methods to include multiple time series in the modeling task is represented by VARIMA models [37,38]. However, despite auto-regressive models exploiting only linear relationships among features, LSTM presents the advantage of leveraging non-linear interactions in the modeling task.

The widespread adoption of LSTM across different domains shows this model’s effectiveness and reliability in multi-step forecasting tasks. Its effectiveness is due to the ability to extract time-variant dependencies and correlations that are inherently present in real-life scenarios and exploiting them to predict future time steps. Different from ARIMA models, which are auto-regressive and capable of analyzing exclusively univariate time series, LSTM models can exploit multiple time series in a combined manner. Potentially, leveraging the existing correlations among them can lead to more accurate predictions. LSTM networks have been widely used for time series multi-step forecasting in multiple studies [32,33,34,35,36].

In [39], a similar approach was described, where convolutional neural networks were combined with LSTM to classify PM10 levels. In [40], an approach for air pollution forecasting using an RNN with LSTM was presented. Multiple models based on deep neural networks have emerged [17,41,42]. Likewise, methods based on fully-connected neural networks such as RNNs [12] and LSTM networks have also emerged. Some approaches combine CNNs to improve the performance of RRN-based air pollution prediction [43,44,45]. In addition, some approaches exploit autoencoder models [46,47], sequence-to-sequence models [48], neural networks that combine linear predictors as ensembles [18], Bayesian networks, and multi-label classifiers [49]. Another interesting approach was explained in [50], where an attention-based model was adopted. This approach’s attention mechanism was applied only to the wind measurements to obtain an encoded value used as a data augmentation technique in the main model. Similarly, in [51], an attention-based approach was applied to all available weather and pollution information. Alternative studies in the literature exploit feature extraction as a preprocessing step for the predictive task [52,53,54,55].

## 3. Methods

The proposed method for pollution forecasting described in this article utilizes the data flow shown in Figure 1. Multiple sensors located at different places in the city provide sensory measurements. Then, several model types are employed to build a forecasting system for predicting the PM10 particle concentration levels in the time horizons of 6, 12, and 24 h.

### 3.1. Dataset and Preprocessing

The dataset consists of air quality sensor measurements from sensors deployed in several locations in Skopje. A variety of parameters are monitored by the sensors, including PM10 and PM2.5, as well as the presence of NO_2_, CO, O_3_, and SO_2_. Measurements were done in intervals of one hour. This dataset was also enriched with meteorological parameters, namely temperature and atmospheric pressure, measured at the Skopje-Petrovec meteorological station. This study evaluated different sliding window lengths, including 6, 12, and 24 consecutive measurements from the air pollution measurement and the meteorological station.

The air pollution measurements were taken hourly, and the meteorological station measurements were taken every three hours. We interpolated the meteorological measurements to obtain the hourly values. The interpolation was performed by repeating the same measurement until a new measurement was obtained.

An earlier study with a similar approach was described in [15], attempting to forecast the PM10 concentration values 3 h in the future. In [39], a similar approach was described, where a subset of the data was used to classify future values using a combination of LSTM and convolutional neural networks. Compared to those approaches, this study also considers the PM2.5 values and concentration of nitrogen dioxide (NO_2_) at the measurement stations in Karposh and Centar municipalities in Skopje, which have not been previously analyzed. The source code repository and the preprocessed dataset are available online (https://gitlab.com/magix.ai/air-pollution-sensors).

Figure 2 shows the seasonality and trend in the data set. It is clearly noticeable that disturbances and irregularities are present in the air quality sensor data. Due to these reasons, to train the recurrent neural network models, we used data in the range from December 2011 to December 2019. To model possible malfunctions of the sensors, we introduced a dropout layer into some of the architectures.

The measurements used are listed below, grouped by location with the longitude and latitude of the locations:Municipality of Karposh, North Macedonia (42.0054876${}^{\circ }$ N, 21.3816153${}^{\circ }$ E)–PM10 concentration–PM2.5 concentration–NO_2_ concentrationMunicipality of Centar, North Macedonia (41.9949946${}^{\circ }$ N, 21.410862${}^{\circ }$ E)–Measurement station Centar—PM10 concentration–Measurement station Centar—PM2.5 concentration–Measurement station Rektorat—PM10 concentrationMunicipality of Miladinovci, North Macedonia (41.9824467${}^{\circ }$ N, 21.6411268${}^{\circ }$ E)–PM10 concentrationMunicipality of Petrovec, North Macedonia (41.9370666${}^{\circ }$ N, 21.6030801${}^{\circ }$ E)–Temperature (in degrees Celsius)–Atmospheric pressure at station level–Daily minimum temperature $T_{min}$Daily maximum temperature $T_{max}$

We performed preprocessing on the sensors’ data and the meteorological data due to different sampling frequencies and time windows with missing values [56]. The preprocessing was performed on the training and validation set, consisting of the following steps:Missing data interpolationMin-Max normalizationTwelve samples’ data window preparation

In addition to the preprocessing, we performed additional reshaping for the CNN layers.

The data were divided into distinct training, validation, and test data sets. For training, we used data in the time interval 1 December 2011–31 December 2019. This dataset consisted of 70,129 samples. Validation samples were taken dynamically as 1 percent from the training data points (709 samples). Before the training process, we used a smaller two year subset of the data for hyperparameter optimization. Data points for the optimization were taken from the interval from 1 August 2014 to 1 August 2016 (17,534 samples). Hyperparameter tuning was validated using a small two month data set in the time frame 1 November 2016–31 December 2016 (1430 samples).

After the models were built and optimized, they were tested on the separate test set, which consisted of measurements from January 2020.

### 3.2. Baseline Model: ARIMA

Among many available time series regression methods, one of the most popular and broadly used is the ARIMA model [57]. Results obtained in this study confirmed that the ARIMA has a strong potential for short-term spot prediction. The ARIMA forms a class of time series models that are widely applicable in time series forecasting. In the ARIMA model, a variable’s future value is a linear combination of past values and errors after removing the trend—by differencing. In our experiments, we adopted the auto ARIMA functionality in order to automatically select the best ARIMA model choosing the optimal values for the number of autoregressive terms (p), the number of nonseasonal differences needed for stationarity (d), and the number of lagged forecast errors in the prediction equation (q). Subsequently, we exploited the best model optimized on the training data for the prediction step.

### 3.3. Deep Learning Models’ Architecture

In this article, we compared several different model architectures and analyzed how they performed compared to the ARIMA model. We used recurrent neural network (LSTM, SimpleRNN) and CNN (Conv2D in combination with MaxPooling2D) layers to build the models. In some of the architectures, we added a dropout layer to mitigate temporary failures of some sensors.

The RNN mainly deals with processing sequence data, such as text, speech, and time series. These types of data exist in an orderly relationship with each other; each piece of data is associated with the previous piece. Another example is climate data, where, for example, the temperature of a day is related to the temperature of the previous day. Therefore, we can form many sets of sequences from a set of continuous data, and the correlation between those sequences can be observed.

We trained all models using the combined data set, using values from the different air quality PM10 measuring stations, and temperature and pressure from the meteorological station. Finally, we validated the ability to make short-term predictions for time horizons of +6, +12, and +24 h in the future. Neural networks are used as a means to reconstruct a function in the form of:(1)$$Y^{n}=f\left(X^{m}\right)$$
where $X^{m}$ is the input vector. Choosing a good-fitting input vector, with appropriate causality and correlation to the desired feature, is not a trivial task. In our experiments, we tried several combinations of features. By providing different sizes of input vectors, we examined which input features contributed the best results. In [15], experiments were done with input vectors ranging from a single PM10 sensor value to a 6 feature vector. Since we were using past values to predict the feature, the input was in the form of a matrix. The shape of the input matrix depends on how many past values will be taken into account. Using models based on LSTM and RNN layers, it was concluded that the best results in short-term air pollution forecasting (+3 h, using a subset of the dataset) can be achieved using the PM10 values from all the sensors in combination with the meteorological parameters. The vector consists of four values of PM10 levels, temperature, and atmospheric pressure. We used these input features as a starting point and examined further possible improvements.

The input for the deep neural networks has the form of the matrix:
$$\left[\begin{array}{ccccc} X_{1,t} & X_{2,t} & X_{3,t} & \ldots & X_{m,t}\\ X_{1,t-1} & X_{2,t-1} & X_{3,t-1} & \ldots & X_{m,t-1}\\ X_{1,t-2} & X_{2,t-2} & X_{3,t-2} & \ldots & X_{m,t-2}\\ \ldots & \ldots & \ldots & \ldots & \ldots \\ X_{1,t-dwl} & X_{2,t-dwl} & X_{3,t-dwl} & \ldots & X_{m,t-dwl} \end{array}\right]$$
where *m* is the number of features and the dwl index represents the data window length. Experiments were done with a data window length of 12 and 24, and the number of features varied in the range from 6 to 9.

We tried to further lower the mean squared error by introducing an extended data set. As a first try, we added two new features to the input. By examining the data seasonality and trend, we added the maximum and minimum daily temperature (T_max, T_min measured at the meteorological station Petrovec) to the training data set. This did not improve the accuracy. Adding too many meteorological parameters seemed to lead to overfitting. Our second attempt was to add the value of the PM2.5 level to the training data set. This value was measured by the same sensor that was used in the forecast. We concluded that the values of the PM10 concentration had a high level of correlation with the level of PM2.5. The extended dataset used in this experiment improved the results. A second value of the measured level of PM2.5 at a different location was added, but it did not bring any further improvement. We also experimented with the measured value of NO_2_, which also is an air pollutant. The LSTM model’s performance decreased when this parameter was used. In most of the experiments, the ReLU function was used as an activation function.
(2)$$ReLU\left(x\right)=x^{+}=max\left(0,x\right)$$
where *x* is the input signal to a neuron. In addition to the ReLU function, experiments were executed to examine the performance of different models using different activation functions like the Scaled Exponential Linear Unit (SELU), tanh, and the sigmoid function.
(3)$$S\left(x\right)=\frac{e^{x}}{e^{x}+1}$$
(4)f(x)=tanh(x)

The SELU is a variant of the ReLU function. It is defined as:(5)SELU(x)=λx
when *x* > 0, or:(6)$$SELU\left(x\right)=\lambda e^{x}-\alpha \lambda$$
when *x* is equal to or smaller than zero. λ and α are pre-defined constants. The SELU solves the problem of vanishing gradients. It was first introduced in [58].

For this particular experiment, we used the mean squared error loss function, and for the model optimization, we used the Adam optimizer [59]. The implementation was done with Keras [60].

### 3.4. Parameter Tuning

We used parameter tuning to obtain the best predictive model. For hyperparameter optimization, a smaller subset of the training data was used. Data points were taken in the interval from 1 August 2014 to 1 August 2016 (17,534 samples). Hyperparameter tuning was validated using a two month validation data set in the time frame 1 November 2016–31 December 2016 (1430 samples). Table 1 presents the parameters that were tuned with the range of values. Optimization was done using the Keras-Tuner library [60] (https://keras-team.github.io/keras-tuner/).

The following parameters ranges were evaluated, aiming to obtain the optimal architecture for this task:Dropout: Deep neural networks with many parameters can be powerful tools. However, overfitting can be a problem in such networks. This often happens when neural nets are trained on relatively small datasets. The lack of control over the learning process often leads to cases where the neural network cannot generalize and make forecasts for new data. Dropout is a technique for addressing this problem. The idea is to randomly drop units from the neural network in the training phase to prevent units from co-adapting too much.Learning rate: The learning rate is a hyperparameter that controls how much to change the model in response to the estimated error each time the model weights are updated. Choosing the learning rate is challenging as a value that is too small may result in a lengthy training process that could get stuck, whereas a value that is too large may result in learning a sub-optimal set of weights too fast or an unstable training process. The learning rate controls how quickly the model is adapted to the problem.LSTM layer units: The number of LSTM cells in the layer is a parameter that we used in our model optimization. The number of units determines the dimensionality of the output space.RNN units: This is number of RNN cells in the layer. By default, the output of an RNN layer contains a single vector per sample. This vector is the RNN cell output corresponding to the last time step, containing information about the entire input sequence. The “units” parameter determines the shape of this output. An RNN layer can also return the entire output sequence for each sample (one vector per time step per sample).Convolutional kernel size: The convolutional kernel size specifies the height and width of the convolution window used as a filter mask in the feature extraction.Number of filters in the Conv2D layer: This is the number of output filters in the convolution.

We performed a grid search through the parameter space, trying every possible combination of the parameters. Table 2 shows which parameters were optimized for each proposed architecture.

## 4. Results

First, we evaluated the impact of the activation function on the model’s performance. Table 3 shows the different MSE and RMSE results obtained with different architectures and activation functions on the validation set. The most optimal activation varied, depending on the model’s architecture. Therefore, in the following experiments, we chose to use the most appropriate activation function for the different architectures, as highlighted in Table 3.

The results for all experiments are presented in Table 4, Table 5 and Table 6. As can be observed, the best result for the 6 h time horizon was obtained using the combination “SimpleRNN + Dense” (MSE = 0.0043, RMSE = 0.0653) . All of the models for this time horizon performed significantly better than the baseline ARIMA model. Figure 3 shows the comparison between the best model and the ARIMA model for the first week of predictions for the six hour time horizon.

For the twelve hour time horizon, most of the proposed approaches were better compared to the baseline ARIMA model (MSE, RMSE): (0.0266, 0.1631). The best model used the “LSTM + Dense” architecture and combined the PM10 measurements with the temperature, pressure, and PM2.5 measures (MSE, RMSE): (0.0151, 0.1227). The results of all models were very close. Figure 4 shows the loss function for the training and validation phases. Figure 5 shows the predictions of this model compared to the ones of the ARIMA model in the first week of the test data set.

For the 24 h time horizon, the best performing approach was the “LSTM + Dropout + LSTM + Dense” architecture, which only slightly outperformed the ARIMA method (MSE = 0.0264, RMSE = 0.1624 vs. MSE = 0.0282, RMSE = 0.1680). Contrary to this, the performance improvement for the 12 h time horizon was significant for the “LSTM + Dense” architecture that used the data of four PM10 stations, temperature and pressure from the meteorological station, and PM2.5 measurements of the target station compared to the ARIMA model (MSE = 0.0151, RMSE = 0.1227 vs. MSE = 0.0266, RMSE = 0.1631). The loss function for the training and validation phases is depicted in Figure 6, and the comparison of the best model for the 24 h horizon can be seen in Figure 7.

As an additional attempt to improve the results, the data set was extended to a window of 24 h of historic data (last two rows of Table 6). This did not bring any significant improvement.

To confirm the above results, we repeated the experiments for the available data from 1 February 2019 to 1 February 2020. The results are presented in Table 7. As can be seen in the results, we came to the same conclusion. The best performing approaches for the period of January 2020 were the same as the best performing approaches on the bigger test set. Unfortunately, due to the large amount of time needed for the ARIMA approach to process the full year of data, we were not able to confirm the performance of the ARIMA model for the 24 h time horizon.

## 5. Discussion

We compared the LSTM-based architecture’s performance with an architecture composed of a SimpleRNN and a dense layer, followed by an experiment combining the SimpleRNN and LSTM layers. The combined architecture did not perform well compared to the simpler architectures where LSTM and SimpleRNN were used separately. In [39], a convolutional architecture was used to classify the value of PM10. Following this approach, we built the model proposed in [39] and used it as a forecasting model. We added an additional step in the data preparation to be able to use this model in forecasting. The CNN provided decent results. It performed better than the SimpleRNN model in forecasting the air pollution 12 and 24 h in advance. To examine the deeper convolutional architecture, a second CNN model was examined, one with an additional Conv2D + MaxPooling2D layer where the number of filters was optimized. The CNN model with three hidden layers did not improve the results. Due to the small number of features, we expected that additional layers could only increase the probability of overfitting on the data. However, further research and experimentation are necessary to prove this.

In addition to this, most of the time, the proposed architectures performed better than the ARIMA model. The exception to this was the combination of the SimpleRNN + LSTM + Dense or SimpleRNN + Dense layer, without dropout, which consistently performed worse than the baseline model. As expected, the proposed architectures’ performance degraded for the longer time horizons for the different models, as shown in Figure 8 and Figure 9 and Table 8. However, the LSTM + Dropout + LSTM + Dense architecture was capable of obtaining a 3.33% RMSE improvement with respect to the ARIMA, which is still a significant result.

The performance degradation might be because the prediction was performed using only a 12 h time window of data as the input. However, even a 24 h time window in the input data did not improve the results. We speculate that the reason for this is that meteorological conditions sometimes drastically change over more extended periods of time (e.g., wind speed, rain, etc.), thus making the predictions much more difficult. Additional experiments are needed with more extended input series to validate that the limitations of the 24 h time horizon prediction result from the shorter input data time periods.

There are several limitations of the study. The first one is that there are no existing sensors for pollution and weather within the same geographical location. Therefore, there is no possibility to include geographical location. Furthermore, there is a limitation in the fact that not all weather parameters were included. The interpolation of the data within the training and the validation set might, in some cases, cover spikes in the pollution, primarily where the interpolation is performed for several hours of data. The obtained results were generated using a training, validation, and test split. Although this method allowed us to get a decent overview of the algorithm’s performance, it is dependent on the random initialization of the variables. We argue that the results were stable by repeating the experiments on a full year of test data and obtaining similar results.

One direction for future work could be an extension of the model to consider derived temporal features, such as month of the year, weekday, holiday, time of day, etc. These features could bring additional information related to traffic and the working hours of factories and industrial plants. To make the analysis more robust and scientifically valid, it would also be preferable to also include data reflecting the traffic intensity to validate that any relations are not random.

## 6. Conclusions

As shown in our previous works, for the Skopje region, the meteorological parameters and measurements are highly correlated with the pollution. When used together with the past data for PM10 and PM2.5 concentrations, they significantly increase the models’ predictive performance. This conclusion is valid even though the measurement stations for pollution and the meteorological measurement stations are several kilometers apart. Researching the geographical location influence within the Skopje city area on the prediction was out of the scope of the work presented in this article. However, further research using more sensory data locations can be used to track how the air pollution changes over time in different city locations. This can also highlight potential causes for the increased pollution.

The proposed architectures performs very well and accurately predicts the short-term concentration of PM10. Compared to the ARIMA baseline model, the short-term predictions are significantly better. As expected, when the time horizon increases, the model’s performance decreases, considering that it is more difficult to make predictions for events further away in the future.

Placing additional meteorological and air quality sensor stations could potentially improve the accuracy of the results. Additionally, it could fine-grain the predictions’ geographical location and allow the authorities to take targeted measures at specific locations instead of citywide measures. There is a rising awareness about the pollution problem within Skopje. Informal sensor stations are already being placed, and their data are being gathered. In future work, we plan to leverage this informal information to include geographical location in our models and possibly improve the current models’ predictive performance.

## Figures and Tables

**Figure 1 sensors-21-01235-f001:**
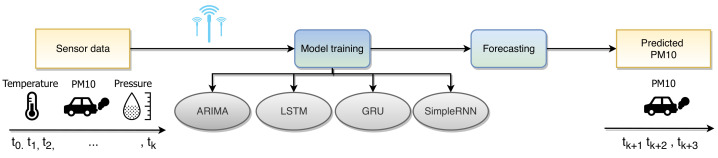
Data flow of the proposed method for air pollution forecasting.

**Figure 2 sensors-21-01235-f002:**
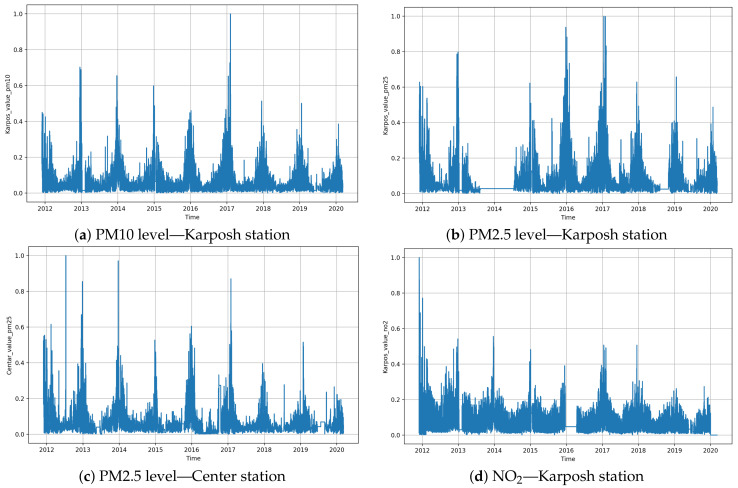
Exploration of the seasonality and trend in concentration levels taken from the Karposh and Center measurement stations.

**Figure 3 sensors-21-01235-f003:**
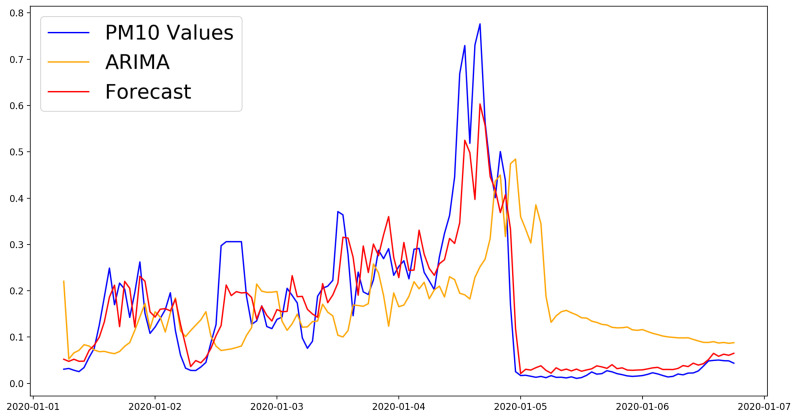
Performance comparison to the ARIMA model of the RNN + Dense model with the best forecasting performance with the 6 h time horizon.

**Figure 4 sensors-21-01235-f004:**
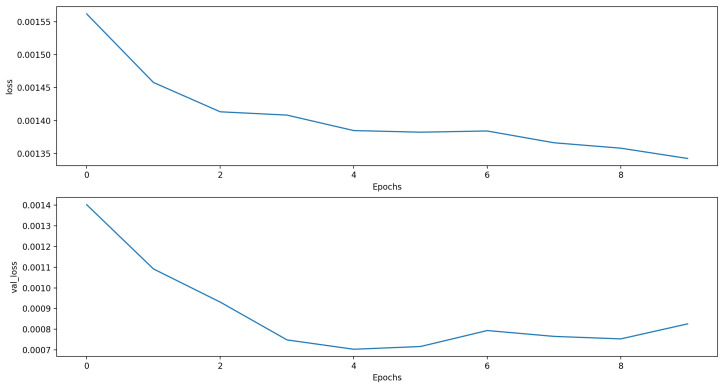
Training and validation MSE of the LSTM + Dense model with the best forecasting performance with the 12 h time horizon. *loss* denotes the training MSE, and *val_loss* represents the validation MSE.

**Figure 5 sensors-21-01235-f005:**
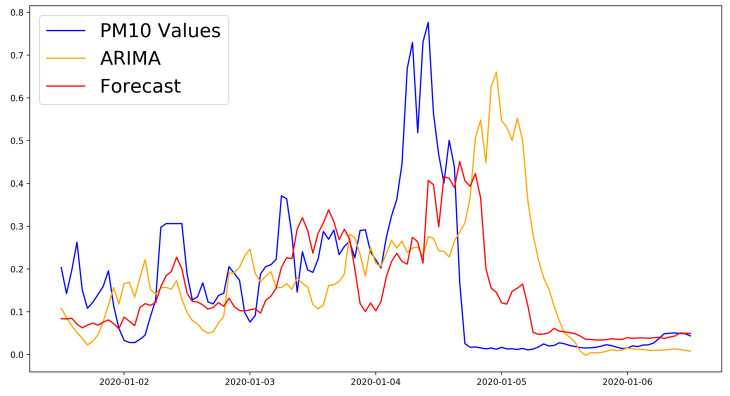
Performance comparison to the ARIMA model of the LSTM + Dense model with the best forecasting performance with the 12 h time horizon.

**Figure 6 sensors-21-01235-f006:**
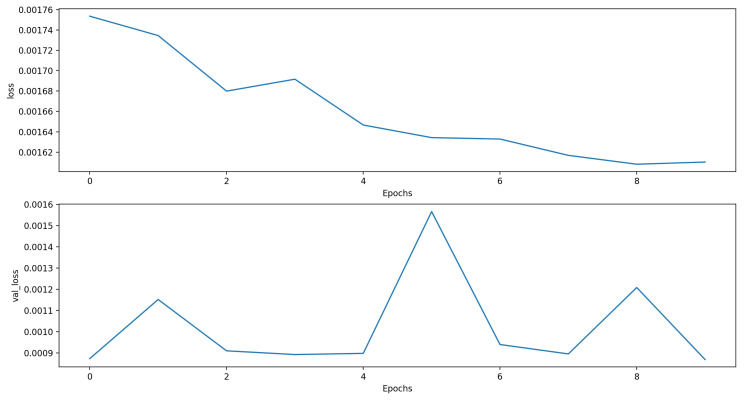
Train and validation MSE of the LSTM + Dense model with the best forecasting performance with the 24 h time horizon.

**Figure 7 sensors-21-01235-f007:**
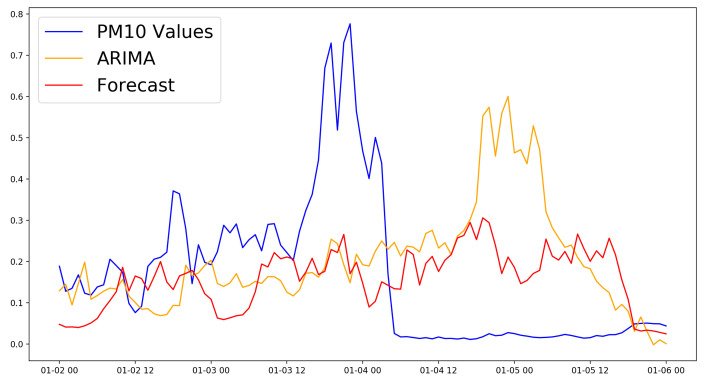
Performance comparison to the ARIMA model of the LSTM + Dense model with the best forecasting performance with the 24 h time horizon in the first week of the test data set.

**Figure 8 sensors-21-01235-f008:**
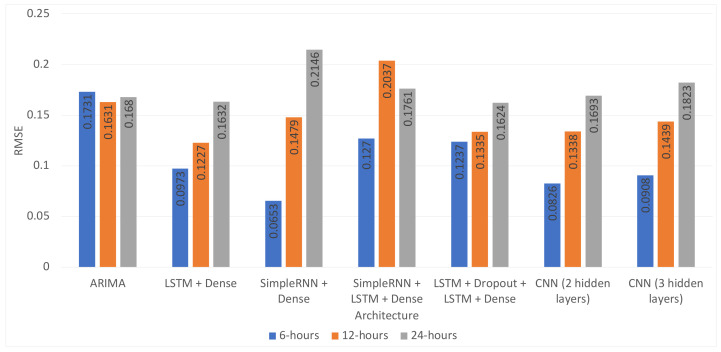
Performance comparison to the best models per architecture and time horizon—one month of test data.

**Figure 9 sensors-21-01235-f009:**
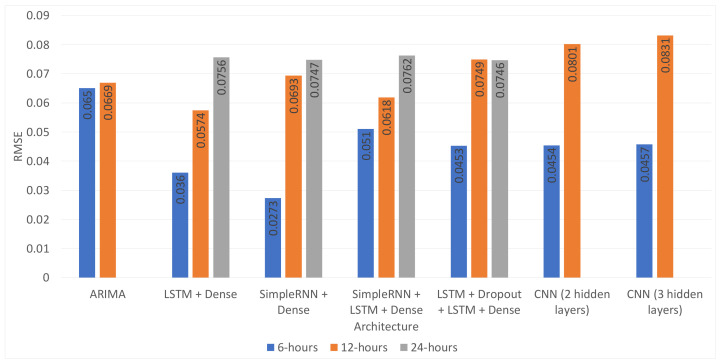
Performance comparison to the best models per architecture and time horizon—one year of test data.

**Table 1 sensors-21-01235-t001:** Parameters used for tuning the neural network.

Parameter Name	Min Value	Max Value	Step
Learning rate	0.01	0.1	[0.01,0.1]
Dropout rate	0.1	0.3	0.1
LSTM 1 layer units	2	128	2
LSTM 2 layer units	24	256	4
RNN 1 layer units	16	256	16
RNN 2 layer units	2	124	4
Kernel size	[3, 3]	[6, 6]	1
Number of filters	4	6	2

**Table 2 sensors-21-01235-t002:** Summary of the evaluated approaches with different configurations in terms of the number of Units in the LSTM layer (U), the Learning Rate (LR), and the Dropout rate (D).

#	Architecture	Parameters Optimized
1	LSTM + Dense	U (2-128) + LR [0.01, 01]
2	ARIMA (12)	None
3	LSTM + Dense	U (2-124) + Learning rates [0.01, 01]
4	LSTM + Dense	U (2-124) + LR [0.01, 01]
5	LSTM + Dense	U (2-124) + LR [0.01, 01]
6	LSTM + Dropout + LSTM + Dense	LSTM (2-24) + D [0.3, 0.2, 0.1]
		+ LSTM (2-124) + LR [0.01, 01]
7	Conv2D + MaxPooling2D + Conv2D + MaxPooling2D	Conv Kernel size ([3, 3]-[6, 6]) + LR [0.01, 01]
	+ Flatten + Dense	
8	Conv2D + MaxPooling2D + Conv2D + MaxPooling2D	Conv. Kernel size ([3, 3]-[6, 6])
	+ Conv2D + MaxPooling2D + Flatten + Dense	+ Third layer filters [4, 6] + LR [0.01, 01]
9	SimpleRNN + LSTM + Dense	RNN (1-128) + LSTM (2-124) + LR [0.01, 01]
10	SimpleRNN + LSTM + Dropout + SimpleRNN + Dense	RNN (1-128) + LSTM (2-24)
		+ D [0.3, 0.2, 0.1] + RNN (2-124) + LR [0.01, 01]
11	SimpleRNN + LSTM + Dropout + LSTM + Dense	RNN (1-128) + LSTM (2-24)
		+ D [0.3, 0.2, 0.1] + LSTM (2-124) + LR [0.01, 01]

**Table 3 sensors-21-01235-t003:** Comparison among different activation functions with the achieved average forecasting performance in terms of Mean Squared Error (MSE) and Root Mean Squared Error (RMSE) for the +6 h time horizon. SELU, Scaled Exponential Linear Unit. The best performance is marked in bold.

Input	Activation Function	Data	Architecture	MSE	RMSE
6 × 12	**ReLU**	4 × PM10 + Temp. + Pressure	SimpleRNN + Dense	**0.0043**	**0.0653**
6 × 12	tanh	4 × PM10 + Temp. + Pressure	SimpleRNN + Dense	0.0055	0.0742
6 × 12	sigmoid	4 × PM10 + Temp. + Pressure	SimpleRNN + Dense	0.0224	0.1497
6 × 12	SELU	4 × PM10 + Temp. + Pressure	SimpleRNN + Dense	0.0076	0.0872
6 × 12	ReLU	4 × PM10 + Temp. + Pressure	LSTM + Dense	0.0166	0.1289
6 × 12	tanh	4 × PM10 + Temp. + Pressure	LSTM + Dense	0.0096	0.0975
6 × 12	**sigmoid**	4 × PM10 + Temp. + Pressure	LSTM + Dense	**0.0095**	**0.0973**
6 × 12	SELU	4 × PM10 + Temp. + Pressure	LSTM + Dense	0.0111	0.1051
6 × 12	ReLU	4 × PM10 + Temp. + Pressure	CNN (3 hidden layers)	0.0128	0.1133
6 × 12	tanh	4 × PM10 + Temp. + Pressure	CNN (3 hidden layers)	0.0106	0.1032
6 × 12	**sigmoid**	4 × PM10 + Temp. + Pressure	CNN (3 hidden layers)	**0.0082**	**0.0908**
6 × 12	SELU	4 × PM10 + Temp. + Pressure	CNN (3 hidden layers)	0.0145	0.1204

**Table 4 sensors-21-01235-t004:** Summary of the evaluated approaches with the achieved average forecasting performance in terms of Mean Squared Error (MSE) and Root Mean Squared Error (RMSE) for the 6 h time horizon. The best performance for is marked in bold.

Input	Hours	Data	Architecture	MSE	RMSE
1 × 12	6	PM10	ARIMA	0.0300	0.1731
6 × 12	6	4 × PM10 + Temp. + Pressure	LSTM + Dense	0.0095	0.0973
6 × 12	6	4 × PM10 + Temp. + Pressure	**SimpleRNN + Dense**	**0.0043**	**0.0653**
6 × 12	6	4 × PM10 + Temp. + Pressure	SimpleRNN + LSTM + Dense	0.0161	0.1270
6 × 12	6	4 × PM10 + Temp. + Pressure	LSTM + Dropout + LSTM + Dense	0.0153	0.1237
6 × 12	6	4 × PM10 + Temp. + Pressure	CNN (2 hidden layers)	0.0068	0.0826
6 × 12	6	4 × PM10 + Temp. + Pressure	CNN (3 hidden layers)	0.0082	0.0908

**Table 5 sensors-21-01235-t005:** Summary of the evaluated approaches with the achieved average forecasting performance in terms of Mean Squared Error (MSE) and Root Mean Squared Error (RMSE) for the 12 h time horizon. The best performance is marked in bold.

Input	Hours	Data	Architecture	MSE	RMSE
1 × 12	12	PM10	ARIMA	0.0266	0.1631
6 × 12	12	4 × PM10 + Temp. + Pressure	LSTM + Dense	0.0155	0.1247
6 × 12	12	4 × PM10 + Temp. + Pressure	SimpleRNN + Dense	0.0219	0.1479
6 × 12	12	4 × PM10 + Temp. + Pressure	CNN (2 hidden layers)	0.0179	0.1338
6 × 12	12	4 × PM10 + Temp. + Pressure	CNN (3 hidden layers) + Conv	0.0207	0.1439
6 × 12	12	4 × PM10 + Temp. + Pressure	SimpleRNN + LSTM + Dense	0.0415	0.2037
7 × 12	12	4 × PM10 + Temp. + Pressure + NO_2_	LSTM + Dense	0.0185	0.1359
7 × 12	12	**4 × PM10 + Temp. + Pressure + PM2.5**	**LSTM + Dense**	**0.0151**	**0.1227**
8 × 12	12	4 × PM10 + Temp. + Pressure + T_max + T_min	LSTM + Dense	0.0166	0.1288
8 × 12	12	4 × PM10 + Temp. + Pressure + T_max + T_min	SimpleRNN + Dense	0.0588	0.2425
8 × 12	12	4 × PM10 + Temp. + Pressure + 2 × PM2.5	LSTM + Dense	0.0156	0.1249
9 × 12	12	4 × PM10 + Temp. + Pressure + PM2.5 + T_max + T_min	LSTM + Dense	0.0161	0.1267
6 × 12	12	4 × PM10 + Temp. + Pressure	SimpleRNN + LSTM + Dropout + SimpleRNN + Dense	0.0178	0.1335

**Table 6 sensors-21-01235-t006:** Summary of the evaluated approaches with the achieved average forecasting performance in terms of the Mean Squared Error (MSE) and Root Mean Squared Error (RMSE) for the 24 h time horizon. The best performance is marked in bold.

Input	Hours	Data	Architecture	MSE	RMSE
1 × 24	24	PM10	ARIMA	0.0282	0.1680
6 × 12	24	4 × PM10 + Temp. + Pressure	LSTM + Dense	0.0266	0.1632
6 × 12	24	4 × PM10 + Temp. + Pressure	SimpleRNN + Dense	0.0461	0.2146
6 × 12	24	**4 × PM10 + Temp. + Pressure**	**LSTM + Dropout + LSTM + Dense**	**0.0264**	**0.1624**
6 × 12	24	4 × PM10 + Temp. + Pressure	CNN (2 hidden layers)	0.0287	0.1693
7 × 12	24	4 × PM10 + Temp. + Pressure + PM2.5 (Karpos)	LSTM + Dense	0.0314	0.1771
8 × 12	24	4 × PM10 + Temp. + Pressure + T_max + T_min	LSTM + Dense	0.0265	0.1628
6 × 24	24	4 × PM10 + Temp. + Pressure	LSTM + Dense	0.0313	0.1769
6 × 24	24	4 × PM10 + Temp. + Pressure	SimpleRNN + LSTM + Dense	0.0310	0.1761

**Table 7 sensors-21-01235-t007:** Summary of the evaluated approaches with the achieved average forecasting performance in terms of Mean Squared Error (MSE) and Root Mean Squared Error (RMSE) for the 6, 12, and 24 h time horizons for the test dataset in the period from 1 February 2019 to 1 February 2020. The best performance for each time horizon is marked in bold.

Input	Hours	Data	Architecture	MSE	RMSE
1 × 6	6	PM10	ARIMA	0.0042	0.0650
6 × 12	6	4 × PM10 + Temp. + Pressure	LSTM + Dropout + LSTM + Dense	0.0021	0.0453
6 × 12	6	4 × PM10 + Temp. + Pressure	SimpleRNN + LSTM + Dense	0.0026	0.0510
6 × 12	6	4 × PM10 + Temp. + Pressure	SimpleRNN + Dense (ReLU)	**0.0007**	**0.0273**
6 × 12	6	4 × PM10 + Temp + Pressure	LSTM + Dense (sigmoid)	0.0013	0.0360
6 × 12	6	4 × PM10 + Temp + Pressure	CNN (sigmoid)	0.0021	0.0454
1 × 12	12	PM10	ARIMA	0.0045	0.0669
6 × 12	12	4 × PM10 + Temp. + Pressure	LSTM + Dense	0.0043	0.0658
8 × 12	12	4 × PM10 + Temp. + Pressure + T_max + T_min	LSTM + Dense	0.0037	0.0606
8 × 12	12	4 × PM10 + Temp. + Pressure + T_max + T_min	SimpleRNN + Dense	0.0048	0.0693
6 × 12	12	4 × PM10 + Temp. + Pressure	CNN (2 hidden layers)	0.0064	0.0801
6 × 12	12	4 × PM10 + Temp. + Pressure	SimpleRNN + LSTM + Dense	0.0038	0.0618
9 × 12	12	4 × PM10 + Temp. + Pressure + PM25 + T_max + T_min	LSTM + Dense	0.0200	0.1414
8 × 12	12	4 × PM10 + Temp. + Pressure + 2 x PM25	LSTM + Dense	**0.0033**	**0.0574**
7 × 12	12	4 × PM10 + Temp. + Pressure + NO_2_	LSTM + Dense	0.0050	0.0705
7 × 12	12	4 × PM10 + Temp. + Pressure + PM25 (Karpos)	LSTM + Dense	0.0034	0.0586
6 × 12	12	4 × PM10 + Temp. + Pressure	SimpleRNN + LSTM + Dropout + SimpleRNN + Dense	0.0056	0.0749
1 × 24	24	PM10	ARIMA	N/A	N/A
8 × 12	24	4 × PM10 + Temp. + Pressure + T_max + T_min	LSTM + Dense	0.0060	0.0773
7 × 12	24	4 × PM10 + Temp. + Pressure + PM25 (Karpos)	LSTM + Dense	0.0060	0.0775
6 × 12	24	4 × PM10 + Temp. + Pressure	CNN (2 hidden layers)	0.1436	0.3790
6 × 12	24	4 × PM10 + Temp. + Pressure	LSTM + Dense	0.0057	0.0758
6 × 12	24	4 × PM10 + Temp. + Pressure	SimpleRNN + Dense	0.0056	0.0747
6 × 12	24	4 × PM10 + Temp. + Pressure	LSTM + Dropout + LSTM + Dense	**0.0056**	**0.0746**
6 × 24	24	4 × PM10 + Temp. + Pressure	LSTM + Dense	0.0057	0.0756
6 × 24	24	4 × PM10 + Temp. + Pressure	SimpleRNN + LSTM + Dense	0.0058	0.0762

**Table 8 sensors-21-01235-t008:** Summary of the evaluated approaches in terms of the percentage of RMSE improvement with respect to the ARIMA for different time horizons. The best performance for each time horizon is marked in bold.

Architecture	6 h	12 h	24 h
LSTM + Dense	43.79	**24.77**	2.86
SimpleRNN + Dense	**62.28**	9.38	−27.74
SimpleRNN + LSTM + Dense	26.63	−24.89	−4.82
LSTM + Dropout + LSTM + Dense	28.54	18.15	**3.33**
CNN (2 hidden layers)	52.28	17.96	−0.77
CNN (3 hidden layers)	47.54	11.77	−8.51

## Data Availability

The source code repository and the preprocessed dataset are available online (https://gitlab.com/magix.ai/air-pollution-sensors).

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
