# Peer review of "Multi-Horizon Air Pollution Forecasting with Deep Neural Networks"

_sensors, 2021, doi:10.3390/s21041235_

Round 1

Reviewer 1 Report

The paper seems to be very UpToDate. It is focused on multi-horizon air pollution forecasting with deep neural networks, that are two trendy topics.

The authors explained the problem in the Introduction. They suppose suggestion deep neural network to forecast multi-horizon pollution. They got data from meteorological stations and combined them with historical date about various pollutants measured in geographically distant places. They decided to evaluate neural networks based on Long-Short Term Memory layers and Convolution neural networks. They used ARIMA for evaluation.

The authors used in the case of both neural networks some parameters to be optimized (dropout, learning rate, number of cells in LSTM layer, cells in RNN layer activation function between two layers and so on). There were used two parameters for recognition of the best result – Mean Square Error and Root Mean Square Error.

The results seem to be correct. They are expressed in tables, figures and numbers. They point out better efficiency of deep neural networks in forecasting air pollution (in comparison with ARIMA).

The result is suggestion of a structure of deep neural network that can forecast air pollution in various location.

Generally, the paper is very interesting. I would like to point out that it could technically understandable even for experts in other fields then deep neural networks. It is methodically clear, and the results are potential.

I suggest publishing it after minor revisions.

I suggest adding these UpToDate sources into literature review (they are focused on pollution, that is on matter, and deep neural networks, that is on methods):

Vochozka, M.; Vrbka, J.; Suler, P. Bankruptcy or Success? The Effective Prediction of a Company’s Financial Development Using LSTM. Sustainability 2020, 12, 7529.

Sheng, P.; Dong, Y.; Vochozka, M. Analysis of Cost-Effective Methods to Reduce Industrial Wastewater Emissions in China. Water 2020, 12, 1600.

Horak, J.; Vrbka, J.; Suler, P. Support Vector Machine Methods and Artificial Neural Networks Used for the Development of Bankruptcy Prediction Models and their Comparison. J. Risk Financial Manag. 2020, 13, 60.

Author Response

Please find the attached revision note with detailed answers to your comments.

Reviewer 2 Report

Referee Report: Multi-horizon air pollution forecasting with deep
neural networks by Mirche Arsov, Eftim Zdravevski, Petre Lameski, Roberto Corizzo, Sasho Gramatikov, Kosta Mitreski, and Vladimir Trajkovik.

The authors describe artificial neural network (RNN, LSTM and CNN) models to predict air pollution from the PM10 pollutant in Skopje, Northern Macedonia. The models are trained on the lagged time series of several pollutants and meteorological variables. Against the baseline of an auto-regressive ARIMA time series model, the relative superiority of the neural network models is demonstrated.

I find the methodology and argumentation convincing in supporting the argument that ANN models are a better performing alternative to classical time series models, and allow to easily train performant prediction models for pollutants. The benefit of reliably predicting short-term pollutant concentrations in metropolitan regions is highlighted in the introduction.

I, however, have a series of technical questions and objections that I would like the authers to address.

1. From Fig. 8, it is apparent -as you state- that for time horizons < 24h the ANN outperform the ARIMA model, which still performs reasonably well. I think it should be explained why at 24h this excess performance vanishes.
2. Given that ARIMA models are much simpler to optimize and to interpret than ANN models, also in terms of feature significance, it should be commented in how far the increase in performance compensates for the loss of interpretability and danger of overfitting, and complexity of hyperparameter tuning.
3. Given that 8 years of training data are available, I find the choice of only one month for restults testing uncompelling. A longer period would allow to obtain more reliable error measures and to better highlight the influence of seasonality on the results.
4. It should be explained what (additional) meteorological and auxiliary data are available, and on what grid and temporal resolution, especially wind velocity and direction. It would also be interesting to include auxiliary variables correlated to traffic intensity - month of year, weekday, and holidays, or, for example, traffic reports.
5. Related to the previous point, a more thorough investigation of seasonality is in order, which can have significant predictive power on its own. I suggest displaying zooms of the training data, together with weekly and monthly averages.
6. It should also be explained why the authors did not use a seasonal ARIMA model (SARIMA) as baseline, and what are the ARIMA parameters.
7. In tables 3-6 and captions/text, please clarify that Input has dimensions m x dwl, and that e.g. Data = 4 x PM10 refers to PM10 measurements at the four sites. Also, please clarify that time horizon refers to the forecasting period as opposed to the lag dwl.
7. Smaller comments:
- l.16b Remove "The" from "The air pollution"
- please verify the format of references 4,5 , and please properly frame the 225Bn USD vs. 5Tn USD numbers in ref. 5 and the text.
- l.94 Yes, but VARIMA can.
- l.132 "disturbances and irregularities" please describe, and explain how they are identified and treated in data preparation. Are the periods with non-functioning sensors excluded from training?
- l. 136 below
- l. 180 this description pf RNN should be stylistically improved and joined with the previous description of LSTM.
- l.188 "implement" -> reconstruct
- l.190 X^m
- l. 205 "Our second attempt was to add the value of PM25 " should be PM2.5
- l. 260 "Be that as it may," is not really informative
- l. 261 "Clearly, the trained model converged and therefore, is very stable." is dubious.
- l.286 "However, further research is experimentation is necessary to
prove this." please check grammar.

Author Response

(The authors gave the same response as above.)

Reviewer 3 Report

The manuscript titled, “Multi-horizon air pollution forecasting with deep neural networks” predicts air pollutants in Skopje, North Macedonia using Recurrent Neural Network (RNN) models relative to more traditional modeling methods such as ARIMA.  Justification for research is that North Macedonia, as a developing country, has a serious problem with air pollution. The problem is highly present in its capital city, Skopje, where air pollution is consistently within the top 10 cities in the world during the winter months.  Authors propose using Recurrent Neural Network (RNN) models with Long Short-Term Memory units to predict the level of PM10 particles at 6, 12, and 24 hours in the future.

Manuscript is well written and organized even though it could be improved by better summary details of validation statistics. 

Comment 1: More detail in Methods section is required on sensors, i.e., the dataset consists of air quality sensor measurements from sensors deployed to several locations in Skopje.  Even though the authors indicate the location by municipality name in North Macedonia.  It would be helpful for the reader to know the geographic distance between the sensors and the latitude and longitude.  How representative are these sensors of overall air pollution levels and why did the authors select the fours sensor locations.  What is the proximity of sensors near auto thoroughfares, manufacturing plants, etc.  

Comment 2: What is the quality of the data from the sensors?  The author state pre-processing was performed on the training and validation sets, consisting of the following steps: Missing data interpolation; Min-Max normalization; 12 samples data window preparation.  This statement in the manuscript suggests some data quality issues.

Comment 3: The results as presented are difficult to compare across the different modeling or algorithm methods.  Presentation of the summary statistics, e.g., RMSE, etc., could be strengthened using RMSEP% given the differences in scale.  The manuscript could be strengthened by using 10-fold cross validation and developing a simple summary table of cross validation comparing the model results as columns and the RMSEP%, etc., for the 10-fold cross validations.  If 10-fold cross validation is not appropriate for this dataset, then ‘Nested Cross Validation’ would strengthen the study.

Comment 4: A sensitivity analysis is possible after more detailed cross or nested-cross validations to analyze the effects in validation as the training sets different, i.e., present some summary statistics (e.g., mean, median, variance, CV) for each training and effect on validation data sets.  The illustrative figures indicate a large difference in scale of pollutants over time.

Comment 5: Summary Figure 8 suggests improvements in short term forecasting (6-hours and 12-hours), discussion on limitations 24-hour predictions should be discussed.

Comment 6: Conclusions should also state the limitations of the study and usefulness of results for practitioners.

Author Response

(The authors gave the same response as above.)

Round 2

Reviewer 3 Report

Paper as revised is acceptable for publication